# Towards Algorithmic Fairness in Space-Time: Filling in Black Holes

**Cheryl Flynn**
AT&T Chief Data Office
Bedminster, NJ, USA
cflynn@research.att.com

**Aritra Guha**
AT&T Chief Data Office
Bedminster, NJ, USA
aritra.guha@att.com

**Subhabrata Majumdar**
Trustworthy ML Initiative
Seattle, WA, USA
zoom.subha@gmail.com

**Divesh Srivastava**
AT&T Chief Data Office
Bedminster, NJ, USA
divesh@research.att.com

**Zhengyi Zhou**
AT&T Chief Data Office
New York, NY, USA
zzhou@research.att.com

## Abstract

New technologies and the availability of geospatial data have drawn attention to spatio-temporal biases present in society. For example: the COVID-19 pandemic highlighted disparities in the availability of broadband service and its role in the digital divide; the environmental justice movement in the United States has raised awareness to health implications for minority populations stemming from historical redlining practices; and studies have found varying quality and coverage in the collection and sharing of open-source geospatial data. Despite the extensive literature on machine learning (ML) fairness, few algorithmic strategies have been proposed to mitigate such biases. In this paper we highlight the unique challenges for quantifying and addressing spatio-temporal biases, through the lens of use cases presented in the scientific literature and media. We envision a roadmap of ML strategies that need to be developed or adapted to quantify and overcome these challenges—including transfer learning, active learning, and reinforcement learning techniques. Further, we discuss the potential role of ML in providing guidance to policy makers on issues related to spatial fairness.

## 1 Motivation

As machine learning (ML) based decision making systems become integrated in our daily lives, demographic fairness and equity concerns continue to surface across diverse applications [33]. Many of these applications are spatio-temporal in nature—such as deployment of public or private services (broadband, 5G, transportation, healthcare, education), or user request prioritization (same-day delivery, rideshare). In spite of the obvious importance, discourse on spatio-temporal applications is largely absent in the extensive literature on algorithmic fairness. Given that the geographical aspect of racial equity and social justice is by now common knowledge—through use cases such as location-based gaming [14], response to COVID-19 [15], and distribution of bikeshare stations [25]—it is important to develop a general methodology for rigorous measurement and mitigation of fairness issues in spatio-temporal data.

Due to the complex dependency patterns that spatio-temporal data exhibits between observations (Section 2), algorithmic fairness techniques designed for non-spatial datasets do not directly carry over into the spatial domain. In this paper, we address this challenge, by outlining the research opportunities and charting the path for future work in developing bias detection and mitigation methods tailored for spatio-temporal data (Section 3). While there are some recent papers that

2022 Trustworthy and Socially Responsible Machine Learning (TSRML 2022) co-located with NeurIPS 2022.

propose fairness-aware methods in the context of urban mobility [49, 50], we lay out the broader vision of how researchers can approach methodological work in the near future that generalizes to a range of use cases and addresses the complex interplay between sources of spatio-temporal demographic bias. Further, we discuss how the results of technical approaches can inform effective policy changes in this context (Section 4). A version of this ongoing work is availble in arXiv which we shall update in future[1].

## 2   Sources of Bias

Based on a literature review of spatio-temporal use cases, we have identified five main sources of demographic bias in this domain.

**Geographical differences:** Geographical differences in residential and/or work areas may cause bias. A large number of developing countries today are located in tropical areas with extreme climatic conditions. Poverty can drastically aggravate the impact on communities living in such areas [13], with land-locked tropical countries typically being among the poorest. Sustainable efforts to countering climate changes also need to ensure the simultaneous upliftment of poverty-stricken areas [43]. Another example of such disproportionate impacts is floods in the United States. The population distribution in floodplains is marked by a disproportionately large number of people from minority communities [37], causing a racial/ethnic bias in determining who gets affected. For example, [20] states that "Of the seven ZIP codes that suffered the costliest flood damage from Katrina, four of them had populations that were at least 75% Black, government records show". Moreover, flooding outside regular floodzones also disproportionately impacts racial and ethnic minorities according to a recent study [44].

**Mental maps** Mental or prejudice maps form another common source of bias. A combination of familiarity bias with spatial homophily in residential clusters [7], along with a historical and continued disproportionate wealth distribution among racial and ethnic minorities [6] means that such communities can be disproportionately affected in a sharing economy. For example, in TaskRabbit, UberX, and most other sharing economy services, low-SES (socio-economic status) individuals have a harder time satisfying microentrepreneur enrollment requirements (e.g., microentrepreneurs must have a bank account in many cases), which likely reduces microentrepreneur participation in low-SES neighborhoods and thereby diminishes the effectiveness of the overall platforms in these neighborhoods. Disproportionate population distribution in low-SES neighbourhoods in turn implies a disproportionate impact on minority communities. Some of these mental maps may be further aggravated by confirmation biases through predictive policing [28] because of differences in crime reporting across population segments, equivalent treatment of unreported crimes with reported ones, and the presence of bias in the initial training data. This further leads to feedback loops [19].

**Spatial clustering:**  Bias can also be induced because of existing residential clustering patterns, where people of different demographic profiles and socioeconomic segments tend to cluster into different spatial regions over time. This has led to uneven access to infrastructure and services, e.g., broadband internet [45], and public transportation [32], and Amazon same-day delivery [23]. Some of these infrastructure and services have spatial constraints on their span and coverage that further worsen the problem of uneven access, e.g., 5G cellular network stations [9] need to be placed close to each other to be within range for communication.

**Spatio-temporal data glitches** Another form of bias occurs from non-random missingness in data. As an example, consider the following. Poverty and wealth information is a factor in deciding government policies. However, if mobile phones are used as the crowdsourcing mechanism [40] to collect that data, it may cause bias in the missingness of the data [15], which may be further aggravated by inferences drawn using the data. Even though almost everyone has a phone, the distribution of the types of phones is not uniform across various population subgroups. For example, it is well-known that iPhone users are more likely to belong to higher income groups than android users. If data collected predominantly through iPhones are used to inform government policies, that would mean a differential impact on lower-income groups, and due to correlation between ethnicity and income, minority groups. Other causes of biased missingness could be lack of access to resources for reporting [8] or context-unaware reuse of external databases. One such example is given by the Pokémon Go example [14]. Data collected as portal locations generated by users

---

[1] http://arxiv.org/abs/2211.04568

of the augmented reality mobile game Ingress were used to determine Gyms and Pokestops for the 2016 game Pokemon Go, and it was later shown that the stops are disproportionately concentrated in White-majority neighborhoods primarily due to the demographic make-up of Ingress players [11]. Other factors that can lead to biased or skewed data collection include legality and actual implementation of collection practices, privacy concerns or preferences, and lack of measurability or charactarizability of certain features (see e.g., [47]).

**Spatio-temporal feedback loop:** We may make decisions today based on biased outcomes resulting from historical practice or demand, forming feedback loops and perpetuating the bias. For example, electric car charging station roll-out [18] prioritizes regions with compatible technologies, products, demand, and willingness to pay. This leads some spatial regions to have less support for electric cars, which reduces demand further for electric cars and subsequently demand for charging stations for these regions. This feedback loop continues to widen the divide across different regions for electric car ownership and its associated environmental impacts. Similarly, "black citizens are about half as likely to live in neighborhoods with access to Amazon same-day delivery as white residents" [23]. Pricing algorithms for ridehailing services such as Uber charge more for neighborhoods with larger non-white populations and higher poverty levels [38]. Historical practices and policies may have also caused such feedback loops. For example, "the use of the [home] sales comparison approach has allowed historical racialized appraisals to influence contemporary values" [22]. Similarly, "[p]eople who live in neighborhoods that were once subjected to a discriminatory lending practice called redlining are today more likely to experience shorter life spans" [21], and poorer cardiovascular health [34]. More recently, discussions around environmental racism in the U.S. and its connection to historical segregation and redlining practices highlight the impact of historical practices and policies. [26] states that "racial/ethnic air pollution exposure disparities persist in part because the underlying sociological, economic, and policy drivers typically evolve on generational time scales."

The five identified sources of bias above often share origins and more than one may impact any particular use case. This can create a confounding effect. Understanding which sources of bias are at play could be useful when identifying mitigation strategies. As pointed out earlier, while geography can play a major role in terms of which areas are affected, social and economic policies and situations can often exacerbate the impact among the poor. Geographical isolation, social exclusion and limited access to education often results in high-levels of poverty among ethnic minorities across the world. Additionally, even when mitigation strategies are implemented, residues of inappropriate racial historical practices such as expropriation of land from Indigenous people, racially exclusive housing covenants, and redlining (in the case of the United States) or racial whitening [17] and restriction of voting to literates (in the case of Brazil) continue to affect minority groups differentially. For example, redlining practices in the United States meant low investments in minority-rich areas, leading to a lack of infrastructure, proper living conditions or access to education. All of these issues have contributed significantly to the five sources of bias identified above.

# 3   Research Opportunities

There are several challenges to bias detection and mitigation in spatial and spatio-temporal settings. First, spatial and spatio-temporal data often exhibit autocorrelation, i.e., neighboring areas and times tend to be similar due to reasons such as residential clustering. As a result, observations may not be independent-and-identically distributed (i.i.d.), as commonly assumed in existing bias detection and mitigation techniques. Second, certain services have additional requirements or constraints in where they should be placed; for example bike sharing stations need to be placed within biking distance to each other. These constraints need to be accounted for or accommodated by any bias detection and mitigation strategies. Third, the confounding effect discussed in the previous section creates challenges for identifying and accounting for the source(s) of bias in spatio-temporal settings. To tackle these challenges, we lay out a roadmap of research opportunities in spatio-temporal bias measurement and mitigation. We discuss three strategies for bias mitigation—gathering additional data or using existing data effectively, algorithmic 'debiasing' methods, and spatial experimentation—that are relevant to the different sources of bias introduced in Section 2 (see Table 1).

Table 1: Promising data-driven mitigation strategies for each source of spatio-temporal bias.

| Source of bias | Mitigation strategies |
|---|---|
| Geographical differences | Data gathering |
| Mental maps | Spatial experimentation |
| Spatial clustering | Algorithmic methods |
| Spatio-temporal data glitches | Data gathering |
| Spatio-temporal feedback loop | Spatial experimentation |

## 3.1 Measurement

Currently, several bias metrics have been proposed in the ML fairness literature [35]; however, these metrics assume independence among data samples. This is often an invalid assumption in spatio-temporal settings and can impact the statistical distribution of a bias metric. For example, [30] demonstrated that ignoring spatial clustering when computing common bias metrics can result in higher false-positive rates than would be expected with independent data samples, and proposed adjusting bias metrics using a spatial filtering approach. Spatial Durbin models have also been used to account for spatial clustering in location-based gaming [14] and the sharing economy [46], where the bias detection problem could be represented by a regression framework. In related work on fair clustering (see [10] and references therein), distance-based bias metrics are commonly used and have been applied in fairness studies on resource center locations. These works focus on detecting a single source of bias, whereas in reality there is often a combination of biases at play. Thus, we identify an interesting research opportunity and two specific research directions.

**Research Opportunity 1.** New algorithmic methods to quantify and detect bias while accounting for the interplay between different sources of spatio-temporal bias.

**Research Direction 1a.** New algorithmic methods to quantify and detect bias while accounting for the impact of underlying human mobility patterns.

For example, resource allocation problems may need to consider human mobility patterns that impact the availability of the resource itself—e.g., Uber drivers—and/or the distribution of individuals with access to the resource—e.g., cell-phone users who move in-and-out of areas with 5G coverage. Bias metrics need to account for patterns in such trajectories, as well as any confounding from mental maps, spatial clustering, or feedback loops that reinforce biased behavior.

**Research Direction 1b.** New algorithmic methods to detect censored feedback tied to varying socio-economic status of geographical areas.

In 2012, the city of Boston launched a smart-phone app to automatically identify potholes by detecting a change in the phone's vertical acceleration and mapping the location using the phone's GPS. By design, pothole reporting was limited to smartphone users, which were predominantly high-income users at that time. Combined with bias in driving behaviors, this resulted in an under-reporting of potholes in lower-income neighborhoods [16]. This example highlights the need for metrics and techniques to automatically detect data glitches due to censored feedback and design choices connected to socio-economic status. One option is to combine data with other cross-referenced sources with related information. For example, the crowd-sourced data for identifying potholes in Boston could have been combined with census or other location data to reveal the lack of uniform coverage across different areas.

## 3.2 Data Gathering

As mentioned in Section 2, bias may arise in the data gathering stage through missing data or skewness of data distributions. These data glitches may also be confounded with other sources of bias such as spatial clustering or a spatio-temporal feedback loop. We propose the following research opportunity and two distinct potential research directions within this space.

**Research Opportunity 2.** New, generalized, or adapted algorithmic methods to mitigate bias caused by data sparsity and quality issues in some spatial regions.

**Research Direction 2a.** Algorithmic approaches to collect more data for under- and misrepresented spatial regions.

To collect more data, active learning strategies [5] can be used to identify from which instances, individuals, regions, or segments to collect data next in the presence of data paucity and resource limitations. Given a fixed budget, we may want to focus on obtaining data from where data is most sparse, uncertain, dissimilar, or biased, and we may need to design additional incentives or targeting to encourage data collection. ML models can also be used in addition to help infer more data or data labels, e.g., semi-supervised learning, cross-referencing with other sources, programmatic label consolidation [41], or data augmentation [27].

**Research Direction 2b.** Algorithmic approaches to increase efficiency of the existing data in the underrepresented spatial regions.

To increase the efficiency of existing data, transfer learning [52] can be used; models can be trained on data from regions or sources with enough well-represented data, and transferred or adapted to regions or sources without. Semi-supervised or weakly supervised models can make use of unlabeled data to learn the underlying data distribution and space.

Publicly available spatial datasets provide useful starting points to evaluate the above opportunities and devise quantitative methods to address them. A few examples include datasets available from Uber [4], the American Community Survey (ACS) data from US Census Bureau [1], and granular city-scale datasets made available by local governments, such as in Chicago [2] and New York [3].

### 3.3 Algorithmic Mitigation

The sizeable recent literature on algorithmic fairness—for non-spatial but related data problems (e.g. social networks)—presents opportunities to design demographic bias mitigation methods for spatial or spatio-temporal problems. To this end, we identify a broad research opportunity with directions for future investigation.

**Research Opportunity 3.** New, generalized, or adapted algorithmic methods to mitigate bias through modifications in different phases of the ML pipeline.

**Research Direction 3a.** Algorithmic approaches to obtain equitable outputs from ML models trained on biased spatial data.

For supervised predictive models, spatio-temporal dependency structure among samples can augment the input dataset (pre-processing), learning objective (in-processing), or model outputs (post-processing). As a pre-processing strategy, input dataset(s) can be replaced with a lower dimensional representation that accounts for fairness constraints *and* underlying spatial autocorrelation [50]. Additonal pre-processing strategies may be explored in the spatio-temporal context. In-processing strategies, such as adversarial debiasing [51] and regularization methods [31], that try to minimize the association between sensitive attributes and predicted outcomes can be generalized to include underlying spatial factors—for example, as a separate regularization term. Post-processing mitigation techniques that aim to minimally change model outputs in a 'fair' way while preserving performance may be adapted to take into account use case-specific spatial constraints.

**Research Direction 3b.** Fair algorithmic methods tailored for geospatial network analysis.

Many domains of geospatial problems—such as transportation and mobility networks or cell phone networks—can be seen as spatial networks, i.e. networks that have nodes and edges embedded in physical space. Relevant directions of future work in our context include adapting fair versions of unsupervised network analysis methods, such as clustering and community detection [12, 48]. The networked nature of discrimination and social (in)equity themselves are well-studied in a non-ML context (e.g. see references in [48]). Such guidance should inform stages in ML workflows such as model choices and data collection strategies when analyzing spatio-temporal data. This is especially important in the face of complexities discussed earlier, such as data glitches and confounding effects.

### 3.4 Spatial Experimentation

As discussed in Section 2, different forms of bias may stem from common sources. In some cases, A/B testing could be used to guide design choices to mitigate underlying biases. For example, awareness campaigns could be evaluated to help address biases [46] in sharing economies by removing false perceptions propagated through mental maps. However, in general, context-unaware usage of data as in [14], or models unaware of spatial associations further perpetuate feedback loops aggravating the biases [18]. In light of this, we identify the following research opportunity and subsequent research directions.

**Research Opportunity 4.** New, generalized, or adapted spatial experimentation methods for bias mitigation.

**Research Direction 4a.** Exploration-exploitation approaches to break feedback loops that further bias.

Reinforcement learning techniques could help balance the exploration/exploitation trade-off. Existing fairness work on the Multi-armed bandit problem [39] and its connections to corporate social responsibility [42] could be adapted to account for the interplay between demographic sub-populations and geographical areas. This would mitigate feedback loops to avoid early adopter biases when expanding existing infrastructure/deploying new products.

**Research Direction 4b.** Context-aware usage of models to overcome existing biases.

Existing literature on context-aware ML models [36] use temporal contexts to determine the best ML model. These works could be extended to consider spatio-temporal contexts to mitigate bias. As an example, assessing the ability for wireless versus wireline broadband technologies to satisfy certain performance criteria is an important task in mitigating broadband deserts [45]. To forecast broadband needs in different neighborhoods, unsupervised learning methods could be used to learn contexts that capture variations in geographical characteristics and socio-economic traits across neighborhoods over time. Then a context-aware ML approach could experiment and determine the best demand forecasting model for each context. The resulting model could be used to determine where different broadband technologies would meet broadband needs.

## 4 Discussions

In this paper, we propose a roadmap for algorithmic methods to detect and mitigate bias in spatio-temporal data settings. We identify five sources of bias that often co-occur and create challenges in spatio-temporal settings, and highlight four main research opportunities to address these challenges.

Translating the research opportunities on algorithmic fairness in space-time identified earlier in this paper to adequately address topics such as climate change, broadband, 5G, transportation, healthcare, and education requires effective policy changes. Given the enormity of the task at hand, a key question that policy makers around the world face is how to prioritize these opportunities.

The importance of data-driven policy making has been widely recognized. As a direct example of how ML practitioners and data scientists can assist policy makers, the RAND corporation recently released an online tool to "...help policymakers and community residents understand the links between historical discriminatory practices and current environmental inequities and identify hot spots of environmental burdens that can be targeted for environmental improvement efforts" [29]. This tool provides a map-based visualization of historically redlined areas compared to different environmental metrics, and box plots summarizing the distributional differences by Home Owners' Loan Corporation (HOLC) zone. To ensure fair policy, the data used for this purpose needs to be comprehensive, unbiased and widely shared. However, as we have seen, available data gathering mechanisms and hence the collected data are often limited. Research directions 2a and 2b will be critical to help policy makers improve the quality and quantity of data available for data-driven policy making.

To this end, the algorithmic techniques identified in the paper to quantify and overcome bias in spatio-temporal data provide a solid technical foundation. Policymakers would be well advised to make effective use of these algorithmic techniques, in conjunction with seeking input from the vari-

ous constituencies that have been historically discriminated against, to create policy in a prioritized manner to maximize their impact. In particular, usage of data or ML models for policies that are context-unaware or unmindful of the spatial associations could lead to undesirable feedback loops. Socioeconomic policies targeted at neighbourhood improvement have the potential to effectively mitigate these biases going forward [24].

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
