# OpenReview forum: "Towards Algorithmic Fairness in Space-Time: Filling in Black Holes"
_NeurIPS.cc/2022/Workshop/TSRML — TSRML2022_

### Official Review · Reviewer_QiXm · 2022-10-20
**Review of #10**

**Overall Rating:** 7

**Summary:**

This paper looks into the fairness problem in spatio-temporal applications which is underexplored but very important in our society. It identifies five unique sources of demographic bias in spatio-temporal use cases and highlights several challenging research problems in this area. This paper provides the fairness community with a good discussion on this critical and practical topic with many future directions.

**Strengths:**

1. This paper sheds light on what and how to study fairness problems in spatio-temporal applications, including the source of biases, measurement, data gathering, algorithms, and experiments.
2. The paper supports its points with many real-world use cases.
3. The paper is well written. It will attract the attention of researchers in the fairness community and inspire future work.

**Weaknesses:**

1. It would be very helpful if authors could suggest some public datasets in the spatio-temporal area that could be used for the studies of fairness.

**Overall Recommendation:**

This is a proposal-like paper. The fairness problems in spatio-temporal area are very important and I believe the discussions in this paper are good contributions to our community.

**Review Confidence:**

3: The reviewer is fairly confident that the evaluation is correct

---

### Official Review · Reviewer_cEuy · 2022-10-21
**Outline of research topics in "algorithmic fairness"**

**Overall Rating:** 6

**Summary:**

The paper presents potential and urgent problems that needs to be tackled in machine learning towards algorithmic fairness with a specific focus on temporal and geographical variability in data.

**Strengths:**

The paper outlines significant and consequential examples of ML applications and highlights various research opportunities. Each topic is further explained on what makes it challenging.

**Weaknesses:**

Even though the paper is a necessary outline, it is neither novel nor it proposes ways to address the challenges indicated. The selection furthermore excludes some topics such as availability of data, legality of collection practices, privacy concerns, or even measurability or charactarizability of certain features (see for example "Fairness for Unobserved Characteristics: Insights from Technological Impacts on Queer Communities").

More importantly, the paper does not mention how one can go about addressing some of these challenges. For instance, context aware machine learning has been a known and deep problem that is tremendously hard to address. Even though topics in this paper are extremely important discussions, some of the challenges have been longstanding and deep problems to solve. An attempt at addressing and highlighting the problems would have been more comprehensive with the inclusion of explicit discussions of bottlenecks and potential roadblocks referring to existing work.



**Overall Recommendation:**

The paper falls short in being constructive yet the examples provided are relevant and the topic needs all the highlights even if they are not uniquely novel to this paper. Therefore, such discussions are tremendously valuable.

**Review Confidence:**

3: The reviewer is fairly confident that the evaluation is correct

---

### Decision · Program_Chairs · 2022-10-23

Accept